# WeDLM: Reconciling Diffusion Language Models with Standard Causal Attention for Fast Inference

**Aiwei Liu** [* 1]   **Minghua He** [* † 2 1]   **Shaoxun Zeng** [3 1]   **Sijun Zhang** [1]   **Linhao Zhang** [1]   **Chuhan Wu** [1]   **Wei Jia** [1]
**Yuan Liu** [1]   **Xiao Zhou** [1]   **Jie Zhou** [1]

 **github.com/tencent/WeDLM**   |    **huggingface.co/collections/tencent/wedlm**

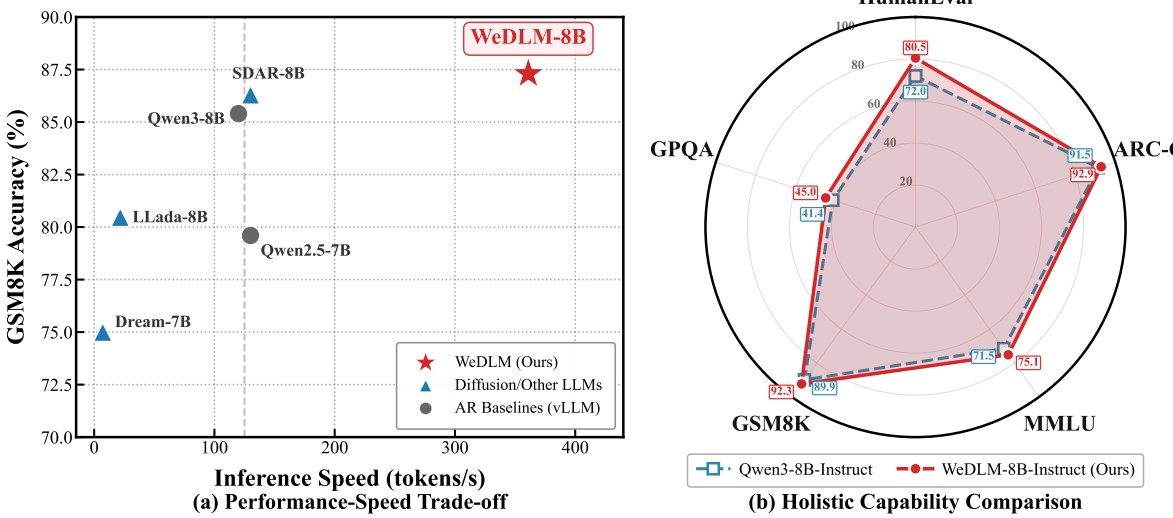

**Figure 1. Performance and capability overview of `WeDLM-8B`.** (a) **Speed vs. Accuracy:** `WeDLM-8B` achieves a ∼3× speedup over the vLLM-optimized AR baseline (`Qwen3-8B`) on GSM8K, while also significantly outperforming prior diffusion models in both inference speed (tps) and accuracy. (b) **Holistic Evaluation:** `WeDLM-8B-Instruct` matches or surpasses the strong capabilities of the `Qwen3-8B-Instruct` baseline, showing improvements across several mathematical, coding, and general knowledge benchmarks.

## Abstract

Autoregressive (AR) generation is the standard decoding paradigm for Large Language Models (LLMs), but its token-by-token nature limits parallelism at inference time. Diffusion Language Models (DLLMs) offer parallel decoding by recovering multiple masked tokens per step, yet in practice often fail to translate this parallelism into speed gains over optimized AR engines (e.g., vLLM)—largely because many DLLMs rely on bidirectional attention, which breaks standard prefix KV caching. We propose **WeDLM**, a diffusion decoding framework built entirely on *standard causal attention* to make parallel generation prefix-cache friendly. The core idea is to let each masked position condition on all observed tokens *while keeping a causal mask*, achieved by *Topological Reordering* that moves observed tokens to the physical prefix while preserving their logical positions. Building on this, we introduce a streaming decoding procedure that continuously commits confident tokens into a growing left-to-right prefix, avoiding the stop-and-wait behavior common in *block diffusion* methods. Experiments show that `WeDLM` preserves the quality of strong AR backbones while delivering substantial speedups, approaching 3× on challenging reasoning benchmarks and up to 10× in low-entropy generation regimes; critically, **our comparisons are against AR baselines served by vLLM under matched deployment settings**.

---
*Equal contribution †Work done during internship at WeChat AI, Tencent. [1]WeChat AI, Tencent, China [2]Peking University, China [3]Tsinghua University, China. Correspondence to: Aiwei Liu <coveliu@tencent.com>.

*Proceedings of the $43^{rd}$ International Conference on Machine Learning*, Seoul, South Korea. PMLR 306, 2026. Copyright 2026 by the author(s).

# 1. Introduction

The autoregressive (AR) generation of Large Language Models (LLMs) is bottlenecked by its step-by-step nature, leaving modern accelerators underutilized in memory-bound regimes (Dao et al., 2022). Diffusion Language Models (DLLMs) offer a compelling trade-off: leveraging more compute per step to generate multiple tokens in parallel (Zhang et al., 2025). Yet despite generating multiple tokens per step, existing DLLMs have not shown clear speed advantages over mature AR serving engines such as vLLM (Kwon et al., 2023). A primary reason is that AR models naturally support prefix caching via KV cache, which, paired with a well-established optimization ecosystem (e.g., PagedAttention (Kwon et al., 2023), CUDA Graphs), delivers substantial speedups. DLLMs, by contrast, struggle to cache prefixes and lack ecosystem support for such acceleration techniques—implying that outperforming AR requires DLLMs to be *prefix-cache compatible*, i.e., they should continuously grow a cache-valid left-to-right prefix so that most computation is reused rather than recomputed.

The core obstacle is the incompatibility between bidirectional attention and prefix caching. Representative DLLMs such as LLaDA (Nie et al., 2025) and Dream (Ye et al., 2025) employ full bidirectional attention, coupling each token's KV state to both past and future positions. Consequently, even resolved tokens cannot be immediately cached—they must be recomputed as later positions change. Block-wise variants (e.g., SDAR (Cheng et al., 2025) and BD3-LMs (Arriola et al., 2025)) partially restore prefix reuse by confining bidirectional attention within each block while keeping inter-block generation autoregressive, enabling block-level KV caching. However, this design has two drawbacks. First, each block requires multiple forward passes, during which early-resolved tokens must wait until the block finalizes before their KV states can be cached. Second, as later diffusion steps resolve fewer remaining positions, the degree of parallelism progressively shrinks within a block. Beyond these, diffusion-style resolution can also proceed out of order, further reducing the fraction of newly predicted tokens that form a contiguous left-to-right prefix eligible for immediate caching. These observations motivate a different design: we argue that bidirectional attention is not essential for parallel mask recovery, and that restoring strict causal structure offers the most direct path to cache-friendly diffusion decoding.

In this work, we propose **WeDLM**, a framework that performs diffusion-style mask recovery entirely under *standard causal attention* to make parallel decoding compatible with prefix caching. Our key insight is that mask recovery only requires each masked position to access all currently observed tokens; this can be achieved without bidirectional attention via *Topological Reordering*. Specifically, we move observed tokens to the physical front while preserving their logical positions through RoPE position ids (Su et al., 2021), so masked tokens can attend to the full observed context under an unmodified causal mask. This causal structure is naturally aligned with prefix caching: once earlier positions are resolved, their KV states depend only on committed context and can be reused immediately. We further introduce *Dual-Stream Masking* to reduce the training–inference gap induced by prefix-conditioned decoding. By constructing a clean *memory stream* alongside a masked *prediction stream* (with shared positional encoding), each prediction block is trained to condition on clean history rather than on potentially noisy intermediate predictions.

For inference, we develop *Streaming Parallel Decoding*, an algorithm explicitly organized around *prefix commitment*. It combines: (i) a position-aware confidence rule, implemented as a distance-penalized selection, that prioritizes earlier unresolved positions and encourages left-to-right growth; (ii) strict causal attention, which guarantees that newly committed prefix tokens become cache-valid immediately; and (iii) a dynamic sliding window that continuously refills new masked slots as soon as tokens are committed, avoiding the stop-and-wait behavior of block-wise methods. Since attention remains a standard causal mask throughout, each iteration reduces to a small causal prefill over the active window on top of an existing KV cache, enabling direct use of optimized AR infrastructure such as FlashAttention (Dao et al., 2022), PagedAttention (Kwon et al., 2023), and CUDA Graphs without kernel modifications.

Experiments show that WeDLM matches or exceeds its base models in capability while achieving significant speedups over optimized AR inference. We instantiate WeDLM on both Qwen2.5-7B (Qwen et al., 2025) and Qwen3-8B (Yang et al., 2025), utilizing 100B tokens for continued pretraining and an additional supervised fine-tuning stage. Across diverse benchmarks, including code generation (MBPP (Austin et al., 2021), HumanEval (Chen, 2021), HumanEval-plus (Liu et al., 2023)), math reasoning (GSM8K (Cobbe et al., 2021), MATH (Hendrycks et al., 2020), GPQA (Rein et al., 2024)), and general knowledge (MMLU (Hendrycks et al., 2021), ARC (Clark et al., 2018), HellaSwag (Zellers et al., 2019)), WeDLM not only preserves but often improves upon the capabilities of its base models. Notably, WeDLM-8B achieves an average score of 77.36 on our benchmark suite, surpassing Qwen3-8B-Instruct (75.12) by over 2 points. **Unlike prior works that compare against unoptimized baselines, we benchmark directly against the state-of-the-art vLLM engine.** Results show up to 3× end-to-end speedup on complex reasoning tasks and over 10× on low-entropy generation; on mathematical reasoning under batched decoding, WeDLM-8B reaches a peak decode throughput beyond 1,400 tokens/s, more than 5× that of the vLLM-served AR baseline at the same batch size,

**demonstrating that a prefix-cache-compatible DLLM can outperform optimized AR engines in practical deployment**.

---

💡 **Main Contributions**

✅ **Causal Diffusion.** We propose `WeDLM`, a DLLM framework that performs mask recovery entirely under causal attention via *Topological Reordering*. This design enables seamless initialization from pretrained AR checkpoints and inherent prefix-cache compatibility—predicted tokens can be cached immediately without waiting for subsequent positions.

✅ **Streaming Parallel Decoding.** We introduce a decoding strategy designed around prefix-cache compatibility: a distance penalty promotes left-to-right resolution, the causal mask enables immediate caching of predicted prefixes, and a dynamic sliding window continuously refills new masks as finalized tokens are committed—eliminating the stop-and-wait bottleneck of block-wise methods.

✅ **First DLLM to Outperform Industrial AR Engines.** We demonstrate that `WeDLM` surpasses vLLM-served AR models in wall-clock speed, achieving over $3\times$ speedups on complex reasoning tasks while maintaining generation quality.

---

**Conflict of Interest Disclosure.** All authors are affiliated with WeChat AI, Tencent. `WeDLM` is developed by the authors at WeChat AI. All baseline models compared in this paper (Qwen2.5, Qwen3, LLaDA, Dream, SDAR) are publicly released by third parties and evaluated using their official checkpoints; the serving baseline vLLM is open-source. The authors declare no other financial conflicts of interest.

## 2. Preliminary

### 2.1. Autoregressive Language Modeling

Given a token sequence $\mathbf{x} = [x_1, \ldots, x_T]$, an autoregressive language model factorizes the joint probability as:

$$P(\mathbf{x}) = \prod_{t=1}^{T} P(x_t \mid x_{<t}; \theta), \quad (1)$$

where $x_{<t} = [x_1, \ldots, x_{t-1}]$ denotes the preceding context.

### 2.2. Decoupled Positional Representation

We make explicit the distinction between a token's *logical position* and its *physical index* by representing inputs as

pairs $(x_t, p_t)$:

$$P(x_t \mid x_{<t}, p_{\leq t}; \theta) = \text{LLM}(x_{\leq t}, p_{\leq t}; \theta). \quad (2)$$

RoPE (Su et al., 2021) naturally supports this decoupling, allowing tokens to be processed in arbitrary physical order while preserving their logical positions.

### 2.3. Masked Diffusion Language Models

MDLMs formulate text generation as denoising. Given a clean sequence $\mathbf{x}_0$ of length $L$, a noising process corrupts a random subset $\mathcal{M}$ (with $|\mathcal{M}| = \gamma L$, $\gamma \in (0, 1]$) by replacing tokens with `[MASK]`, yielding $\mathbf{x}_\gamma$. The model is trained via weighted cross-entropy:

$$-\mathbb{E}_{\gamma, \mathbf{x}_0, \mathbf{x}_\gamma} \left[ \frac{1}{\gamma} \sum_{i=1}^{L} \mathbf{1}[x_\gamma^{(i)} = \mathbf{M}] \log p_\theta(x_0^{(i)} \mid \mathbf{x}_\gamma) \right], \quad (3)$$

where $1/\gamma$ compensates for varying mask counts.

## 3. Motivation and Analysis

Two observations shape `WeDLM`'s design: (1) In KV-cached deployment, decoding speed is governed by *prefix cacheability* rather than per-step parallelism. (2) Mask recovery does not require bidirectional attention; it can be implemented with *standard causal attention*.

### 3.1. Prefix Cacheability ($p_{\text{cache}}$) as an Inference Metric

Prior DLLMs mostly pursue speed by increasing *tokens predicted per forward*. In practice, a more critical factor is how many predicted tokens can form a *growing, KV-cache-valid prefix*. With KV caching, a token is reusable only if its key/value states depend solely on earlier context; thus only a left-to-right prefix is cacheable. Tokens outside the committed prefix must be recomputed in later forwards.

We quantify this with two indicators. Let $N_{\text{gen}}$ be the number of new tokens produced (excluding prefill), and $N_{\text{fwd}}$ the total token instances processed across all decoding forwards (including recomputation). The *prefix cacheability* is

$$p_{\text{cache}} \triangleq \frac{N_{\text{gen}}}{N_{\text{fwd}}} \quad \in (0, 1], \quad (4)$$

interpreted as the probability that a processed token instance becomes final and cache-reusable. The average recomputation factor is $1/p_{\text{cache}}$.

This metric captures an efficiency dimension distinct from per-step parallelism. Fully bidirectional methods (e.g., LLaDA, Dream) may predict many tokens per forward, yet achieve low $p_{\text{cache}}$ because few predictions are immediately cache-valid. Block-wise methods (e.g., SDAR, NBDiff) improve speed largely by increasing $p_{\text{cache}}$ via partial prefix commitment. Improving $p_{\text{cache}}$ can match or exceed

speedups from increasing per-step parallelism. (See Appendix F for detailed comparisons.)

## 3.2. Rethinking the Necessity of Bidirectional Attention

Standard MDLMs (Nie et al., 2025; Ye et al., 2025) adopt bidirectional attention so that each position can aggregate information from all others. While natural, this is *not* a requirement of the mask-recovery objective itself. Our key observation is that the information flow needed for mask recovery can be realized under *standard causal attention* by enforcing two principles:

> ### *Design Principles for Causal Mask Recovery*
>
> $\mathcal{O}$ denote observed positions and $\mathcal{M}$ masked positions.
>
> (i) **Observed-Context Visibility:** Each masked position should attend to all observed tokens $x_{\mathcal{O}}$.
>
> (ii) **Directed Dependence Among Masks:** We impose a directed ordering over $\mathcal{M}$ so that each masked position attends only to preceding masked positions, replacing symmetric visibility with a causal structure.

Principle (i) captures the essential requirement of MDLM-style denoising. Principle (ii) is a modeling choice: we parameterize dependencies within the masked set using a causal factorization $q_\theta(x_{\mathcal{M}} \mid x_{\mathcal{O}}; \pi) = \prod_{j=1}^{|\mathcal{M}|} q_\theta(x_{\pi(j)} \mid x_{\mathcal{O}}, x_{\pi(<j)})$ under an ordering $\pi$, allowing earlier-resolved masked tokens to influence later ones without bidirectional attention.

We emphasize that we do *not* claim equivalence to bidirectional-attention MDLMs: under our design, clean tokens still follow their own causal ordering, so the resulting attention values differ from fully bidirectional attention. What our two principles guarantee is that every masked token can attend to all observed tokens, which is the only conditioning required by masked diffusion denoising. The ablation in Figure 5(c) further shows that causal intra-block attention in fact outperforms its bidirectional counterpart for AR-initialized models, indicating that full bidirectionality is not necessary in this regime. Whether the directed dependence among masks is sufficient in practice is otherwise evaluated empirically in §5.

Combining §3.1 and the above, our goal is to maintain *standard causal attention* while ensuring each masked position accesses the *full observed context*. We achieve this via *Topological Reordering* (§4.1) and address the training–inference gap using *Dual-Stream Masking* (§4.2).

## 4. WeDLM: Training and Inference

This section presents WeDLM's framework, which reconciles parallel decoding with *standard causal attention*. We introduce *Topological Reordering* (§4.1) to expose all observed tokens to masked positions under an unmodified causal mask, *Dual-Stream Masking* (§4.2) to mitigate the training–inference mismatch induced by prefix-conditioned decoding, and *Streaming Parallel Decoding* (§4.3) for efficient inference with immediate KV commitment.

### 4.1. Causal Mask Recovery via Topological Reordering

To ensure masked positions can attend to all observed tokens under standard causal masking, we introduce *Topological Reordering*. We apply a permutation that places all observed tokens before masked tokens in the *physical computation order*, while decoupling this from *logical text positions* (indexed by position embeddings). This allows every masked token to access the full observed context using unmodified causal attention.

**Problem Setup.** Consider a clean sequence $\mathbf{x}_0 = [x_1, x_2, \ldots, x_L]$ with logical positions $\mathbf{p} = [1, 2, \ldots, L]$. We sample a masking ratio $\gamma \in (0, 1]$ and uniformly select indices $\mathcal{M} \subset \{1, \ldots, L\}$ with $|\mathcal{M}| = \gamma L$ to be masked. The remaining indices $\mathcal{O} = \{1, \ldots, L\} \setminus \mathcal{M}$ are observed, with $|\mathcal{O}| = N_o$ and $|\mathcal{M}| = N_m$.

**Topological Reordering.** We construct a reordered sequence $\tilde{\mathbf{x}}$ by placing observed tokens before masked tokens:

$$\tilde{\mathbf{x}} = [\underbrace{x_{o_1}, x_{o_2}, \ldots, x_{o_{N_o}}}_{\text{observed tokens}}, \underbrace{\texttt{[M]}, \texttt{[M]}, \ldots, \texttt{[M]}}_{N_m \text{ mask tokens}}], \quad (5)$$

where $\{o_1, \ldots, o_{N_o}\}$ are observed indices sorted ascendingly and $\texttt{[M]}$ is a shared mask token. Logical positions are preserved via:

$$\tilde{\mathbf{p}} = [\underbrace{o_1, o_2, \ldots, o_{N_o}}_{\mathbf{P}_o}, \underbrace{m_1, m_2, \ldots, m_{N_m}}_{\mathbf{P}_m}], \quad (6)$$

where $\{m_1, \ldots, m_{N_m}\}$ are masked indices, also sorted ascendingly.

**Context Awareness under Causal Masking.** Under causal attention, a token at physical index $i$ attends only to indices $\{1, \ldots, i-1\}$. In $\tilde{\mathbf{x}}$, observed tokens occupy $\{1, \ldots, N_o\}$ and masked tokens occupy $\{N_o + 1, \ldots, L\}$; thus every masked token attends to *all* observed tokens. Positional encodings (e.g., RoPE) are indexed by $\tilde{\mathbf{p}}$, so attention scores reflect logical relative offsets.

**Training Objective.** With $(\tilde{\mathbf{x}}, \tilde{\mathbf{p}})$, we train the model to recover ground-truth tokens at masked positions. Following Eq. 2:

$$-\mathbb{E}_{\gamma, \mathbf{x}_0, \mathcal{M}} \left[ \frac{1}{\gamma} \sum_{j=1}^{N_m} \log P_\theta \left( x_0^{(m_j)} \mid \tilde{\mathbf{x}}_{<N_o+j}, \tilde{\mathbf{p}}_{<N_o+j} \right) \right], \quad (7)$$

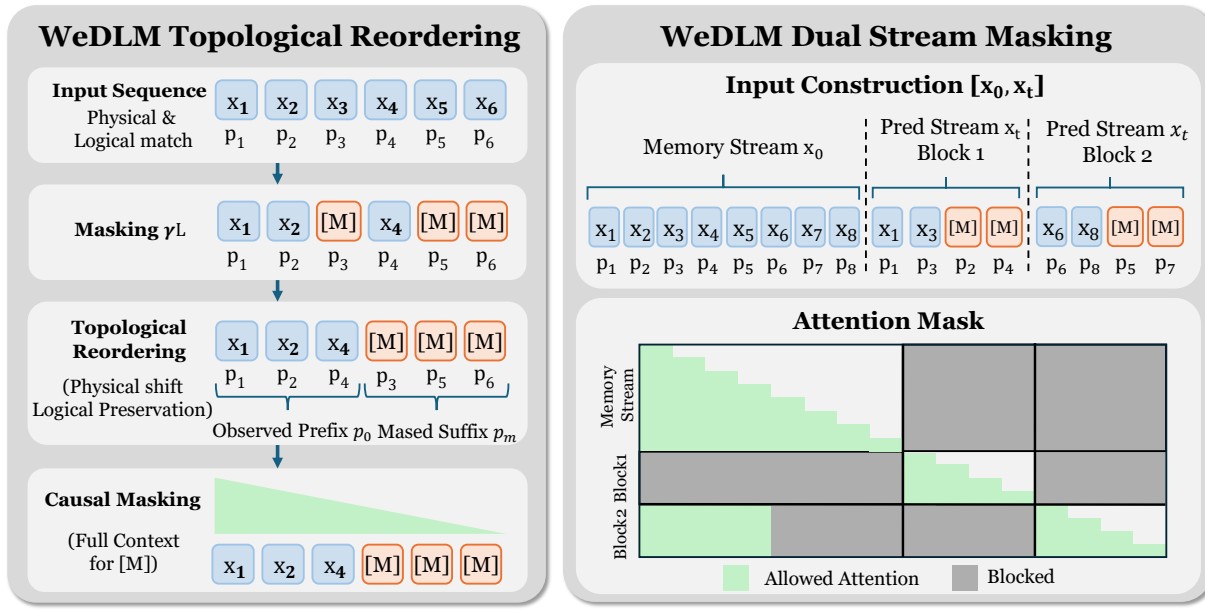

*Figure 2.* **Overview of the `WeDLM` training framework. Left:** *Topological Reordering* physically shifts observed tokens to the prefix while preserving logical positions. This grants masked tokens access to the full observed context under standard causal masking. **Right:** *Dual-Stream Masking* concatenates a clean Memory Stream with a masked Prediction Stream. The block-wise attention mask ensures that the Prediction Stream conditions on clean memory history rather than noisy preceding predictions, aligning training dynamics with inference.

where $m_j$ is the logical position of the $j$-th masked to-ken and the factor $1/\gamma$ follows Eq. 3. Unlike bidirectional MDLMs, we operate under strictly causal attention: each masked token conditions only on earlier *physical* positions, yet accesses the full observed context through topological reordering.

### 4.2. Dual-Stream Masking for Training

The objective in Eq. 7 masks tokens uniformly over the sequence. During inference, however, unresolved tokens predominantly reside in a (block-wise) suffix due to left-to-right progression, inducing a train–inference distribution gap (Cheng et al., 2025; Tian et al., 2025). A naive fix—masking only short suffixes—would exclude most tokens from the loss. We therefore propose *Dual-Stream Masking*, which simulates suffix-style decoding while preserving training efficiency.

Given a clean sequence $\mathbf{x}_0 = [x_1, x_2, \ldots, x_L]$ with positions $\mathbf{p} = [1, 2, \ldots, L]$, we construct a *memory stream* $\mathbf{x}_o$ and a *prediction stream* $\mathbf{x}_t$, both initially identical to $\mathbf{x}_0$, concatenated as:

$$\mathbf{x}_{\text{input}} = [\quad \underbrace{\mathbf{x}_o}_{\text{memory stream}}, \quad \underbrace{\mathbf{x}_t}_{\text{prediction stream}}]. \quad (8)$$

Critically, both streams share the same position sequence $\mathbf{p}_{\text{input}} = [1, \ldots, L, 1, \ldots, L]$, placing them in the same positional reference frame (e.g., under RoPE). The two streams are distinguished by their physical segment and the attention mask.

We partition the prediction stream $\mathbf{x}_t$ into $K = \lceil L/B \rceil$ non-overlapping blocks of size $B$. For each block $k$, we sample a masking ratio $\gamma_k \in (0, 1]$ and apply *intra-block* topological reordering as in §4.1: observed tokens move to the front and masked tokens to the back, while logical positions are preserved. The memory stream $\mathbf{x}_o$ remains unmasked and unreordered.

The attention mask matches inference-time conditioning: for a token in block $k$ of $\mathbf{x}_t$, its visible context comprises (1) all memory tokens from $\mathbf{x}_o$ whose logical positions precede block $k$, and (2) tokens within block $k$ of $\mathbf{x}_t$ that precede it in the reordered physical sequence. Notably, block $k$ cannot attend to previous blocks within $\mathbf{x}_t$; it accesses clean history from $\mathbf{x}_o$ instead, simulating inference where earlier blocks are finalized.

Let $\mathcal{M}_k$ denote masked positions within block $k$. We aggregate losses across blocks:

$$-\mathbb{E}_{\{\gamma_k\}, \mathbf{x}_0} \left[ \sum_{k=1}^{K} \frac{1}{\gamma_k} \sum_{j \in \mathcal{M}_k} \log P_\theta \left( x_0^{(j)} \mid \mathbf{x}_o^{(<k)}, \tilde{\mathbf{x}}_t^{(k, <j)} \right) \right],$$

where $\mathbf{x}_o^{(<k)}$ denotes memory tokens preceding block $k$, and $\tilde{\mathbf{x}}_t^{(k,<j)}$ denotes tokens in reordered block $k$ physically preceding position $j$.

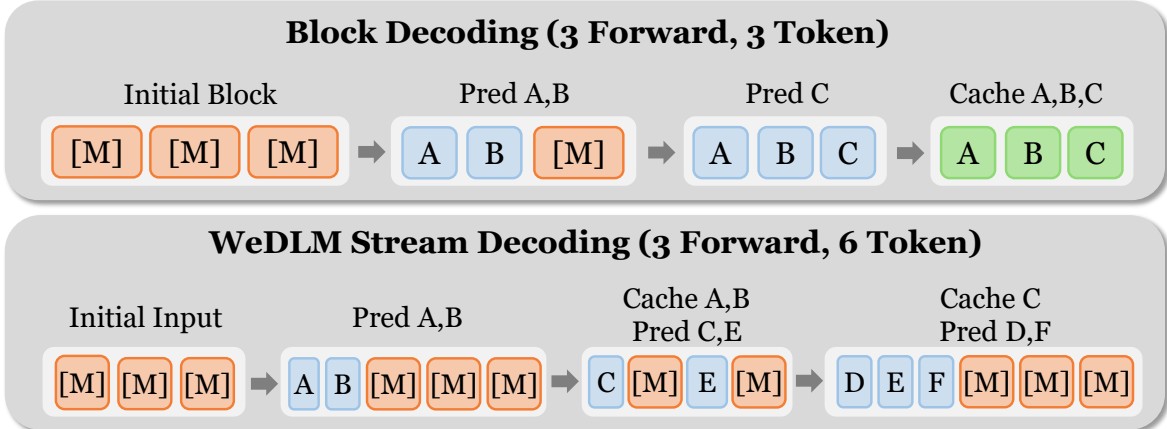

*Figure 3.* **Block Decoding vs. WeDLM Streaming Parallel Decoding.** Block decoding suffers from stop-and-wait: bidirectional dependence prevents committing any token until the entire block is finalized. In contrast, WeDLM uses standard causal attention with a dynamic sliding window: resolved tokens (e.g., A, B) are immediately committed, while new mask tokens (e.g., C, E) are appended for parallel prediction.

### 4.3. Streaming Parallel Decoding

In KV-cached serving, latency depends not only on tokens proposed per forward, but also on how effectively these proposals can be *committed into a contiguous left-to-right prefix*. Efficient streaming decoding requires: (i) *immediate KV validity*—a predicted token's representation should depend only on committed context to avoid recomputation; (ii) *left-to-right commitment*—prioritizing earlier positions to extend the cacheable prefix; and (iii) *no stop-and-wait*—maintaining steady workload without block-boundary stalls. WeDLM's causal attention naturally satisfies (i), while our decoding strategy addresses (ii)–(iii).

**Procedure.** We maintain a window of $W$ slots containing filled (predicted but uncommitted) and [M] (pending) tokens. Each step: **(a)** reorder so filled tokens precede masks (positions preserved via global ids), **(b)** run causal forward on the persistent cache, **(c)** commit the leftmost contiguous filled prefix into cache, **(d)** predict additional masks by confidence and position, **(e)** refill with new masks. The full algorithm is provided in Algorithm 1 in Appendix.

**Position-Aware Confidence.** To increase the chance that resolved tokens form a contiguous prefix, we bias selection toward earlier positions. Following Ye et al. (2025), we use prediction entropy for confidence. Let $p_i(\cdot)$ be the predicted distribution at mask slot $i$ with entropy $H_i$. We define a distance-adjusted entropy:

$$\tilde{H}_i = H_i + \lambda \cdot d_i, \qquad (9)$$

where $d_i$ is the distance from slot $i$ to the leftmost remaining mask, and $\lambda > 0$ controls left-to-right preference. Masks with $\tilde{H}_i < \tau$ are filled with their argmax predictions.

Under causal masking, leftmost filled slots attend only to

the cache and earlier filled slots, making them immediately cache-valid. Committed tokens are removed and new [M] slots appended, keeping work per forward constant. The procedure is natively supported by FlashAttention, PagedAttention (Kwon et al., 2023), and CUDA Graphs.

## 5. Experiments

We evaluate WeDLM on generation quality (§5.3), inference efficiency (§5.4), and conduct ablation studies (§5.5).

### 5.1. Training Details

We initialize WeDLM from Qwen2.5-7B (Qwen et al., 2025) and Qwen3-8B (Yang et al., 2025). We perform continued pretraining on 100B tokens with learning rate from $3 \times 10^{-6}$ to $3 \times 10^{-7}$ following a cosine schedule. For Dual-Stream Masking (§4.2), we set block size $B = 32$. To handle irregular attention patterns from topological reordering, we employ Magi Attention (Zewei & Yunpeng, 2025), which accelerates computation over non-rectangular masks without custom CUDA kernels.

To preserve autoregressive capabilities, we incorporate an auxiliary AR loss computed on the same sequences using standard next-token prediction. After pretraining, we perform supervised fine-tuning (SFT) on 10K internal instruction-response pairs with learning rate $3 \times 10^{-6}$ and cosine decay. The resulting models are denoted WeDLM-7B and WeDLM-8B.

### 5.2. Evaluation Setup

We evaluate on benchmarks spanning reasoning, knowledge, and code generation: ARC-Challenge (Clark et al.,

*Table 1.* Main results on generation quality across diverse benchmarks for **Instruct models**. We compare our WeDLM against autoregressive (AR) baselines and recent diffusion language models (DLLMs). The columns for our model are highlighted in blue. Best results in each row are **bolded**.

| Benchmark | AR Baseline *Instruct Model* | | DLLM Baseline *Instruct Model* | | | WeDLM (Ours) *Instruct Model* | |
|---|---|---|---|---|---|---|---|
| | Qwen2.5-7B | Qwen3-8B | LLaDA-8B | Dream-7B | SDAR-8B | **WeDLM-7B** | **WeDLM-8B** |
| *General Reasoning* | | | | | | | |
| ARC-C (0-shot) | 86.09 | 91.47 | 85.92 | 87.20 | 91.13 | 89.59 | **92.92** |
| ARC-E (0-shot) | 93.27 | 96.17 | 94.32 | 93.27 | 97.01 | 96.09 | **97.43** |
| HellaSwag (10-shot) | 87.59 | 86.13 | 78.55 | 62.00 | **92.12** | 84.75 | 82.94 |
| MMLU (5-shot) | 71.98 | 71.52 | 63.70 | 64.19 | 73.61 | 70.52 | **75.14** |
| *Math & Science* | | | | | | | |
| GSM8K (3-shot) | 89.91 | 89.91 | 80.59 | 79.00 | 91.66 | 87.57 | **92.27** |
| MATH (4-shot) | 45.00 | **69.60** | 34.20 | 41.00 | 43.40 | 55.40 | 64.80 |
| GPQA-Diamond (5-shot) | 27.10 | 41.41 | 25.25 | 35.86 | 38.38 | 33.84 | **44.95** |
| *Code Generation* | | | | | | | |
| MBPP (3-shot) | 63.66 | 68.37 | 36.24 | 58.52 | 67.97 | 63.66 | **70.53** |
| HumanEval (4-shot) | 76.22 | 71.95 | 36.59 | 57.32 | 76.83 | 75.00 | **80.49** |
| HumanEval-plus (4-shot) | 70.12 | 64.63 | 32.32 | 51.22 | 70.12 | 71.34 | **73.78** |
| **Average** | 71.09 | 75.12 | 56.77 | 62.96 | 74.22 | 72.78 | **77.53** |

2018) (0-shot), GPQA (Rein et al., 2024) (5-shot), HellaSwag (Zellers et al., 2019) (10-shot), MMLU (Hendrycks et al., 2021) (5-shot), GSM8K (Cobbe et al., 2021) (3-shot), MATH (Hendrycks et al., 2020) (4-shot), MBPP (Austin et al., 2021) (3-shot), and HumanEval (Chen, 2021) (4-shot). For generative tasks (GSM8K, GPQA, MBPP, HumanEval, MATH), we set maximum generation length to 512 tokens with sampling temperature 0.1.

For fair comparison, results in Tables 2 and 1 use a unified step-wise decoding scheme: each step generates one token by selecting the lowest-entropy position; we use window size $W = 6$ and distance penalty $\lambda = 0.10$ (Eq. 9). We compare against autoregressive baselines (Qwen2.5-7B, Qwen3-8B) and diffusion models: LLaDA-8B (Nie et al., 2025), Dream-7B (Ye et al., 2025), and SDAR-8B (Cheng et al., 2025). Each model uses its recommended inference framework: LLaDA and Dream use dInfer, SDAR uses JetEngine, Qwen and WeDLM use vLLM (Kwon et al., 2023), demonstrating seamless compatibility with industrial inference systems.

### 5.3. Performance Evaluation

Tables 2 (in Appendix) and 1 report generation quality under *base* and *instruct* settings. Across both, WeDLM preserves and often improves upon its AR checkpoints while maintaining a large margin over prior diffusion models.

On **base models**, WeDLM-7B achieves 70.84 average (+3.6 over Qwen2.5-7B) and WeDLM-8B reaches 74.72 (+2.1

over Qwen3-8B). Gains concentrate on reasoning tasks: GSM8K (+5.5 and +4.2), MATH (+4.8 and +2.8), and GPQA-Diamond (+3.2 and +5.4). HumanEval shows notable gains (+9.8 for 7B, +6.1 for 8B), while MBPP is the only benchmark with consistent drops (3–4 points).

On **instruct models** (Table 1), WeDLM-8B achieves the strongest results at 77.53 average (+2.4 over Qwen3-8B), with consistent gains on MMLU (+3.6), GPQA-Diamond (+3.5), HumanEval (+8.5), and HumanEval-plus (+9.2). These results indicate that diffusion-style training does not conflict with instruction tuning, and can amplify it from strong instruct checkpoints.

Compared to **diffusion baselines**, WeDLM maintains clear advantages. On base models, LLaDA-8B and Dream-7B average 55–57, which is 15–19 points below WeDLM. On instruct models, the best diffusion baseline (SDAR-8B) averages 74.22 versus WeDLM-8B's 77.53.

### 5.4. Speed Evaluation

Figure 4 examines key inference hyperparameters using WeDLM-7B-Instruct. The entropy threshold $\tau$ (Figure 4a) controls unmasking confidence: performance remains stable (∼53–54%) for $\tau \leq 0.5$, then degrades sharply as low-confidence predictions propagate errors. We recommend $\tau \in [0.3, 0.6]$. The distance penalty $\lambda$ (Figure 4b) biases selection toward left-positioned tokens, directly increasing $p_{\text{cache}}$ by forming contiguous committed prefixes. Increasing $\lambda$ from 0.01 to 0.05 improves accuracy by 2.6

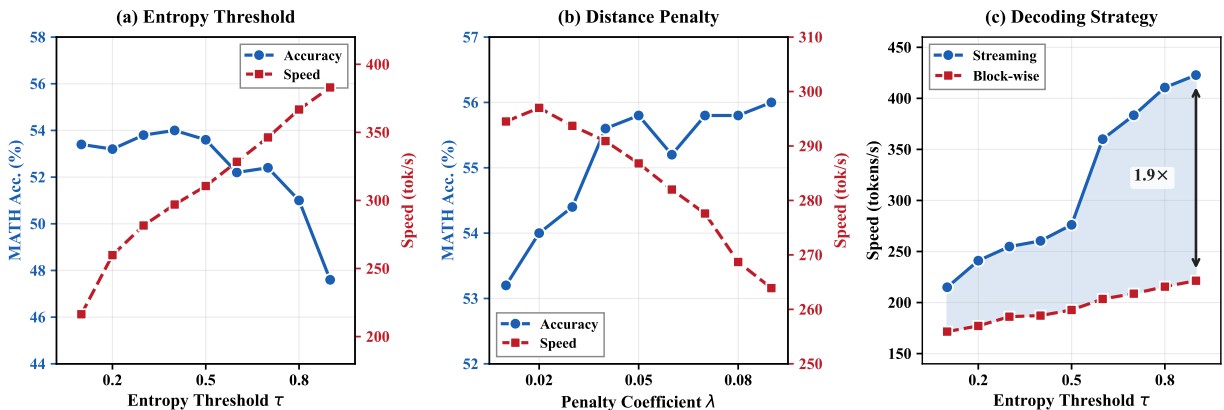

*Figure 4.* Ablation studies on inference hyperparameters. (a) Effect of entropy threshold $\tau$ on MATH accuracy and speed, with optimal range $\tau \in [0.3, 0.5]$. (b) Effect of distance penalty $\lambda$: prioritizing left-positioned tokens improves accuracy with minimal speed cost. (c) Streaming vs. block-wise decoding: streaming achieves up to $1.9\times$ speedup by enabling immediate prefix commitment.

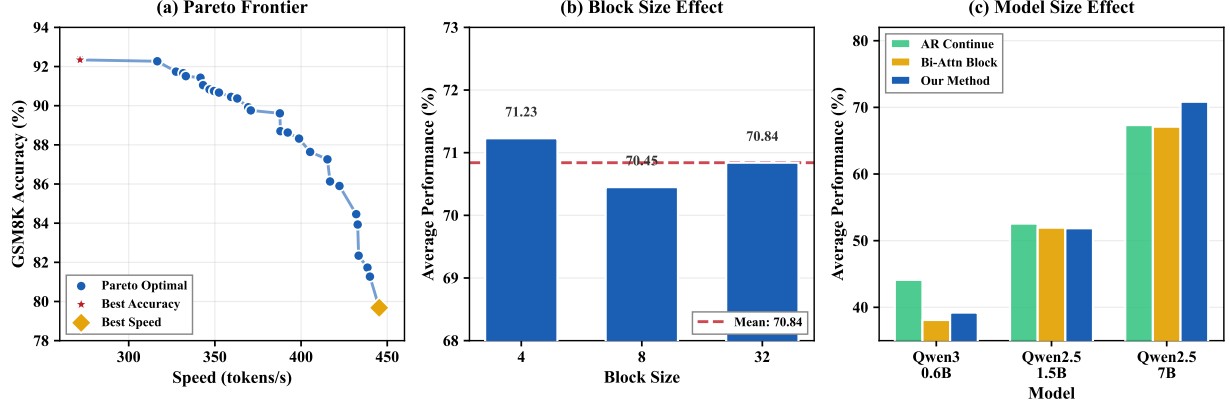

*Figure 5.* **Ablation studies.** (a) Pareto frontier on GSM8K: conservative settings achieve 92.3% accuracy at $1.97\times$ speedup; aggressive settings reach $3.2\times$ acceleration. (b) Block size effect: stable performance across $B \in \{4, 8, 32\}$. (c) Attention design and scale: causal attention outperforms bidirectional intra-block attention; larger models benefit more from causal adaptation.

points with only 3% speed reduction, confirming that left-to-right resolution accelerates caching and reduces error accumulation.

Figure 4(c) shows Streaming Parallel Decoding consistently outperforms block-wise decoding. At $\tau = 0.9$, streaming achieves $1.9\times$ speedup (423 vs. 221 tokens/s). Block-wise methods must wait until entire blocks finalize before tokens become cache-valid, yielding lower $p_{\text{cache}}$; streaming commits tokens immediately upon forming contiguous prefixes.

Figure 5(a) presents the Pareto frontier on GSM8K using `WeDLM-8B-Instruct`, spanning 79.7–92.3% accuracy at 272–445 tokens/s. Conservative settings ($\tau = 0.2$) preserve near-baseline accuracy at $1.97\times$ speedup, while aggressive settings ($\tau = 0.9$) achieve $3.2\times$ acceleration above 79% accuracy. Qualitatively, decoding speed correlates strongly with the entropy of the output distribution: counting, structured math derivations, and open-ended explanations span over $8\times$ in achievable throughput. Representative examples spanning low-, medium-, and high-entropy regimes are provided in Appendix F.1.

## 5.5. Ablation Studies

Figure 5(b) examines block size $B$ during continued pretraining. Performance remains virtually identical across $B \in \{4, 8, 32\}$ (within 0.8 points), demonstrating insensitivity to block size. This favors larger blocks in practice: Magi Attention incurs higher overhead for smaller blocks, and models trained with larger $B$ support any smaller window at inference without retraining. Concretely, the inference window size $W$ can be *freely adjusted at deployment time without any re-training*: a model trained with $B = 32$ runs equally well with $W \in \{4, 8, 16, 32\}$, letting practitioners trade off parallel speedup against per-step latency or memory at serve time.

Figure 5(c) compares our method against two baselines: (1) continued AR pretraining on the same data, and (2) bidirectional attention *within* blocks. Our causal intra-block design achieves higher average performance than both alternatives, indicating that directed factorization suffices for AR-initialized models and that the gains stem from the diffusion-based parallel decoding rather than merely additional train-

ing. Moreover, bidirectional intra-block attention limits $p_{\text{cache}}$—tokens cannot commit until entire blocks resolve—whereas causal design enables immediate per-token caching.

Figure 5(c) also shows that model scale affects adaptation: smaller models (0.6B, 1.5B) experience slight degradation compared to AR continue training, while 7B shows consistent improvement (+3.5 points over AR continue). The improvement correlates with base model capability, suggesting 7B+ models are recommended for WeDLM deployment.

## 6. Related Work

**Discrete Diffusion Language Models.** Discrete diffusion models enable parallel generation through iterative denoising. RADD (Ou et al., 2024) simplified the framework with time-independent concrete scores, while Nie et al. (2024) established scaling laws showing masked diffusion requires $\sim 16\times$ more compute to match AR perplexity. LLaDA (Nie et al., 2025) first scaled masked diffusion to 8B parameters, and LLaDA-MoE (Zhu et al., 2025) matched dense models with 1/6 active parameters. Recent work enhances diffusion LMs through reinforcement learning (Zhao et al., 2025; Wang et al., 2025a; Pan et al., 2025). Given investment in pretrained AR models, DiffuLLaMA (Gong et al., 2024) introduced shift operations and attention mask annealing for AR-to-diffusion adaptation, while Dream 7B (Ye et al., 2025) achieved strong performance with only 0.6T tokens via context-adaptive noise rescheduling. Dream-Coder (Xie et al., 2025) extended this to code generation.

**Block Diffusion and Inference Acceleration.** Block diffusion applies diffusion within fixed-size blocks while maintaining AR dependencies across blocks. BD3-LM (Arriola et al., 2025) introduced vectorized training and clipped noise schedules; NBDiff (Tian et al., 2025) proposed gradual block growth; SDAR (Cheng et al., 2025) demonstrated lightweight adaptation with dynamic truncation; SDLM (Liu et al., 2025) introduced adaptive speculative decoding; SBD (Gat et al., 2025) unified next-token and masked-token prediction; and LLaDA2.0 (Bie et al., 2025) scaled block diffusion to 100B parameters.

**KV-Caching for Bidirectional DLLMs.** A complementary line of work retrofits caching onto *bidirectional*-attention dLLMs: Fast-dLLM (Wu et al., 2025) approximately reuses KV states across diffusion steps for LLaDA/Dream, while D2F (Wang et al., 2025b) distills a block-wise causal structure into a bidirectional model. WeDLM instead adopts strict causal attention from the start via Topological Reordering, so these post-hoc approximations are not needed. Efficient-DLM (Fu et al., 2025) is the closest prior block-diffusion approach—also adapting AR models with block-wise attention—but keeps *bidirectional* attention within each block and thus inherits stop-and-wait behavior,

whereas WeDLM's fully causal design enables *immediate per-token commitment*.

**Permutation and Reordering.** XLNet (Yang et al., 2019) studies permutation language modeling, i.e., training an autoregressive objective under random factorization orders with permutation-dependent masking via two-stream attention to avoid information leakage. WeDLM differs in both goal and mechanism: we focus on *inference-time* acceleration with diffusion-style parallel decoding while *keeping standard causal attention*. Our Topological Reordering simply moves currently observed tokens to the physical prefix so masked tokens can attend to them under an unmodified lower-triangular mask, preserving logical positions (e.g., via RoPE position ids) and remaining KV-cache friendly.

## 7. Conclusion

We introduced WeDLM, a diffusion-style decoding framework explicitly optimized for *prefix-cacheable* generation under standard causal attention. Our core insight is that, in KV-cached decoding, the dominant efficiency driver is not "tokens predicted per forward" but the *rate at which predictions become a contiguous prefix*—formalized via $p_{\text{cache}}$ (Eq. 4)—explaining why out-of-order resolution and bidirectional KV coupling are fundamentally misaligned with fast decoding. WeDLM addresses this through Topological Reordering, which exposes full observed context while preserving the causal mask, and Streaming Parallel Decoding, which biases acceptance toward earlier positions and continuously refills a fixed window for sustained prefix growth. Empirically, WeDLM retains the capabilities of strong AR backbones while delivering substantial inference acceleration. More broadly, our results argue that *prefix-cacheability should be a first-class design objective* for parallel text generation: generating many tokens per iteration is only beneficial insofar as those tokens can be quickly promoted into a cache-valid prefix under a causal computation graph.

**Limitations and Future Work.** The price of prefix-cache compatibility is reduced post-hoc editability: once a token is committed into the KV cache it cannot be revised, although tokens within the uncommitted streaming window can still be refined across steps. Our empirical evidence is concentrated on 7B/8B dense models initialized from strong AR checkpoints; extending WeDLM to larger dense and MoE backbones, and to training a causal dLLM *from scratch*, are natural next steps.

## Acknowledgements

We sincerely thank the anonymous reviewers WsdG, KYUv, and 9ae3, as well as the Area Chair and Senior Area Chairs, for their careful reading and constructive feedback during the review process. Their questions on training-token com-

parisons, batch-throughput evaluation, the relationship to bidirectional-attention dLLMs, and the precise positioning of `WeDLM` with respect to Fast-dLLM, D2F, and Efficient-DLM have meaningfully improved the clarity and the empirical scope of this paper.

## Impact Statement

This paper presents work that aims to advance the field of Machine Learning. There are many potential societal consequences of our work, none of which we feel must be specifically highlighted here.

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

*Table 2.* Main results on generation quality across diverse benchmarks for **Base models**. We compare our `WeDLM` against autoregressive (AR) baselines and recent diffusion language models (DLLMs). The columns for our model are highlighted in blue. Best results in each row are **bolded**.

| Benchmark | AR Baseline *Base Model* | | DLLM Baseline *Base Model* | | WeDLM (Ours) *Base Model* | |
|---|---|---|---|---|---|---|
| | Qwen2.5-7B | Qwen3-8B | LLaDA-8B | Dream-7B | `WeDLM-7B` | `WeDLM-8B` |
| *General Reasoning* | | | | | | |
| ARC-C (0-shot) | 89.93 | 92.66 | 81.14 | 88.40 | 90.70 | **92.92** |
| ARC-E (0-shot) | 96.55 | 97.13 | 92.00 | 96.21 | 96.13 | **97.14** |
| HellaSwag (10-shot) | 80.20 | 85.27 | **85.34** | 78.05 | 85.11 | 84.55 |
| MMLU (5-shot) | 71.62 | 74.03 | 64.61 | 70.64 | 71.93 | **75.46** |
| *Math & Science* | | | | | | |
| GSM8K (3-shot) | 79.23 | 85.97 | 71.80 | 75.97 | 84.76 | **90.20** |
| MATH (4-shot) | 43.40 | 50.80 | 28.00 | 38.00 | 48.20 | **53.60** |
| GPQA-Diamond (5-shot) | 33.70 | 37.00 | 29.80 | 25.76 | 36.87 | **42.42** |
| *Code Generation* | | | | | | |
| MBPP (3-shot) | 65.30 | **70.94** | 41.99 | 56.47 | 61.81 | 67.00 |
| HumanEval (4-shot) | 59.14 | 68.90 | 31.71 | 20.12 | 68.90 | **75.00** |
| HumanEval-plus (4-shot) | 53.05 | 63.40 | 28.05 | 19.51 | 64.00 | **68.90** |
| Average | 67.21 | 72.61 | 55.44 | 56.91 | 70.84 | **74.72** |

## A. Streaming Parallel Decoding Algorithm

Algorithm 1 presents the complete procedure for Streaming Parallel Decoding introduced in §4.3. The algorithm maintains a fixed-size sliding window $\mathcal{W}$ of length $W$ that contains both filled tokens (predictions from previous iterations) and mask tokens awaiting prediction. Each iteration performs four key operations: (1) *Reorder & Forward* applies topological reordering to place filled tokens before masks, then executes a causal forward pass over the reordered window; (2) *Commit* identifies the leftmost contiguous prefix of filled tokens, appends them to the output sequence $\mathbf{y}$, and extends the KV cache $(\mathbf{K}, \mathbf{V})$ with their key-value states; (3) *Predict* selects a subset of mask positions based on position-aware confidence (Eq. 9) and fills them with sampled predictions; (4) *Refill* appends new mask tokens to maintain constant window size and parallelism. The procedure terminates when all positions are resolved.

The algorithm's efficiency stems from three design choices. First, strict causal attention ensures that once a token is committed to the prefix, its KV states depend only on earlier context and require no recomputation. Second, the position-aware confidence mechanism (via distance penalty $\lambda$) biases selection toward earlier positions, increasing the likelihood that resolved tokens form a contiguous prefix eligible for immediate commitment. Third, the dynamic refilling strategy maintains a constant degree of parallelism across iterations, avoiding the progressive shrinkage observed in block-wise methods. Together, these properties enable `WeDLM` to achieve high prefix cacheability ($p_{\text{cache}}$) while remaining fully compatible with standard serving infrastructure such as vLLM, FlashAttention, and PagedAttention.

## B. Training Cost

The continued pretraining of `WeDLM-8B` processes 100B tokens using 256 GPUs over approximately 5 days, totaling ∼30,720 GPU-hours. While dual-stream masking doubles the sequence length and introduces additional computational overhead, we require significantly less training data compared to standard AR pretraining, making the overall cost impact limited. The topological reordering primarily affects data preprocessing rather than the training loop itself.

**Training tokens vs. other dLLMs.** A practical advantage of starting from a strong pretrained AR backbone is data efficiency. `WeDLM` reaches its reported quality with only **100B** tokens of continued pretraining, whereas LLaDA-8B (Nie

---

**Algorithm 1** Streaming Parallel Decoding

---

1: **Input:** Prompt prefix $\mathbf{x}$, window size $W$, entropy threshold $\tau$, distance penalty $\lambda$
2: **Output:** Generated sequence $\mathbf{y}$
3: $\mathbf{y} \leftarrow []$; $(\mathbf{K}, \mathbf{V}) \leftarrow \textsc{Prefill}(\mathbf{x})$
4: $\mathcal{W} \leftarrow [[\texttt{M}]]^W$ {Each slot carries a fixed global position id}
5: **while** $\mathcal{W} \neq \emptyset$ **do**
6:     {**Reorder & Forward**: filled tokens placed before masks}
7:     $\mathcal{W} \leftarrow [\mathcal{W}_{\text{filled}}; \mathcal{W}_{\text{mask}}]$
8:     $(\boldsymbol{\ell}, \mathbf{K}_{\mathcal{W}}, \mathbf{V}_{\mathcal{W}}) \leftarrow \textsc{Forward}(\mathcal{W}, \mathbf{K}, \mathbf{V})$
9:     {**Commit**: commit the leftmost contiguous filled prefix}
10:     $n \leftarrow \min\{i : \mathcal{W}[i] = [\texttt{M}]\}$ or $|\mathcal{W}|$ if none
11:     Append $\mathcal{W}[0{:}n]$ to $\mathbf{y}$
12:     Extend $(\mathbf{K}, \mathbf{V})$ with $(\mathbf{K}_{\mathcal{W}}[0{:}n], \mathbf{V}_{\mathcal{W}}[0{:}n])$
13:     $\mathcal{W} \leftarrow \mathcal{W}[n{:}]$
14:     {**Predict**: fill a subset of masks based on confidence}
15:     $\mathcal{F} \leftarrow \textsc{SelectByEntropy}(\boldsymbol{\ell}_{\text{mask}}, \tau, \lambda)$
16:     **for all** $i \in \mathcal{F}$ **do**
17:         $\mathcal{W}[i] \leftarrow \textsc{Sample}(\boldsymbol{\ell}_i)$
18:     **end for**
19:     {**Refill**: append new masks to maintain constant parallelism}
20:     $\mathcal{W} \leftarrow [\mathcal{W}; [[\texttt{M}]]^n]$
21: **end while**
22: **return** $\mathbf{y}$

---

et al., 2025) is trained from scratch on **2.3T** tokens and Dream-7B (Ye et al., 2025) uses roughly **1.3T** tokens of diffusion pretraining. `WeDLM` therefore uses on the order of $\sim 4\%$ of the tokens of LLaDA-8B and $\sim 8\%$ of Dream-7B while matching or exceeding their generation quality. This concretizes the "continued pretraining" framing used throughout the paper: the architectural gains do not require pretraining-from-scratch budgets.

**Wall-clock with irregular masks.** The block-wise attention masks induced by topological reordering and dual-stream masking are non-rectangular and therefore not directly supported by the stock FlashAttention (Dao et al., 2022) kernel. We instead use MagiAttention (Zewei & Yunpeng, 2025), which handles heterogeneous mask patterns efficiently without writing custom CUDA kernels. With this substitution, the per-step wall-clock cost of `WeDLM` training (at sequence length 4096) is roughly $\sim 1.3\times$ that of a standard causal AR pass at the same sequence length, with per-GPU memory at $\sim 1.5\times$; coupled with the $\sim 10\times$ reduction in training tokens above, the end-to-end cost remains an order of magnitude below pretraining a comparable bidirectional dLLM from scratch.

## C. Additional Experimental Results

Table 2 presents comprehensive evaluation results for base models without instruction tuning. `WeDLM-7B` and `WeDLM-8B` demonstrate consistent improvements over their respective AR initialization checkpoints (Qwen2.5-7B and Qwen3-8B) across most benchmarks, with particularly strong gains on mathematical reasoning (GSM8K, MATH) and code generation (HumanEval, HumanEval-plus). Compared to prior diffusion baselines (LLaDA-8B, Dream-7B), `WeDLM` maintains a substantial margin of 14–18 points in average performance, confirming that causal attention with topological reordering preserves model capability while enabling efficient parallel decoding. The results validate that diffusion-style continued pretraining does not degrade base model quality and can enhance performance on reasoning-intensive tasks.

## D. Batch Throughput across Task Types

The speed numbers in the main text focus on single-request decoding, which most closely matches latency-sensitive deployments. For completeness, we also evaluate `WeDLM-8B-Instruct` in a batched serving setting against the same Qwen3-8B baseline running under vLLM. We measure decode-only throughput (excluding prefill) on three representative task categories—mathematical reasoning, code generation, and open-ended QA—across batch sizes $\{1, 8, 16\}$, after warmup,

averaged over three runs.

Decoding in modern LLM serving is overwhelmingly *memory-bandwidth bound*: every generated token requires reloading the full model weights from device memory, while the arithmetic units stay largely idle. Predicting multiple mask tokens within the same forward pass is therefore close to "free" on the compute side, which is the same principle that lets batched AR serving improve throughput nearly linearly. WeDLM exploits this idle compute by issuing several mask predictions per forward; the predicted tokens are immediately eligible for prefix commitment thanks to the causal attention design.

*Table 3.* Decode-only throughput of WeDLM-8B-Instruct vs. Qwen3-8B (vLLM) under different batch sizes, broken down by task type. "DLM tok/fwd" is the average number of tokens WeDLM commits per forward pass.

| Task | BS | AR (tok/s) | DLM (tok/s) | Speedup | DLM tok/fwd |
|------|-----|-----------|-------------|---------|-------------|
| Math | 1 | 110.2 | 686.5 | 6.2× | 6.58 |
| Math | 8 | 243.6 | 1475.8 | 6.1× | 7.00 |
| Math | 16 | 258.3 | 1389.5 | 5.4× | 6.26 |
| Code | 1 | 109.2 | 223.8 | 2.1× | 2.11 |
| Code | 8 | 243.1 | 539.6 | 2.2× | 2.50 |
| Code | 16 | 256.4 | 610.3 | 2.4× | 2.68 |
| QA | 1 | 107.6 | 162.5 | 1.5× | 1.57 |
| QA | 8 | 239.2 | 324.9 | 1.4× | 1.51 |
| QA | 16 | 254.6 | 321.3 | 1.3× | 1.42 |

*Table 4.* Throughput aggregated across the three task categories. WeDLM is never slower than vLLM-served AR across the tested batch sizes; the speedup decreases mildly as the system transitions from memory-bound to compute-bound.

| Batch Size | AR (tok/s) | DLM (tok/s) | Speedup |
|------------|-----------|-------------|---------|
| 1 | 109.0 | 357.6 | 3.3× |
| 4 | 213.5 | 680.8 | 3.2× |
| 8 | 241.9 | 780.1 | 3.2× |
| 16 | 256.4 | 773.7 | 3.0× |

WeDLM is *never slower* than the vLLM-served AR baseline across all tested batch sizes and task types, with an aggregated speedup of $3.0\times$–$3.3\times$ and up to $5.4\times$–$6.2\times$ on math. As batch size grows, the system gradually shifts from memory-bound to compute-bound, leaving less idle compute to amortize across mask predictions; this is the expected reason that the aggregated speedup drops mildly from $3.3\times$ at BS=1 to $3.0\times$ at BS=16, and confirms our explanation that WeDLM's acceleration originates from filling otherwise-idle compute during memory-bound decode.

## E. Ablation on Key Components

We validate the effectiveness of two critical design components: Dual-Stream Masking and Auxiliary AR Loss. Figure 6 presents ablation results on MATH, GSM8K, HumanEval, and MBPP benchmarks.

**Dual-Stream Masking.** Without the dual-stream structure (using only uniform masking), performance drops by 5–10 points across all tasks (average: 7.9 points). This confirms that simulating the inference-time distribution—where unresolved tokens concentrate in a suffix—is essential for effective training.

**Auxiliary AR Loss.** Removing the auxiliary autoregressive objective results in consistent 2–3 point drops (average: 2.6 points). This validates our design choice to maintain autoregressive capabilities alongside diffusion-style training.

## F. Prefix Cacheability Metric

Table 5 compares the prefix cacheability metric $p_{\text{cache}}$ across diffusion models on different task categories. All models are evaluated with block size $B = 8$ for parallel generation. We report results on four representative benchmarks: MATH and GSM8K (mathematical reasoning), HumanEval and MBPP (code generation).

**WeDLM exhibits consistent $p_{\text{cache}}$ across tasks.** As shown in Table 5, WeDLM achieves $p_{\text{cache}} \in [0.45, 0.48]$ across all four benchmarks, with only 0.03 variance. This stability stems from our causal attention design: the streaming parallel decoding

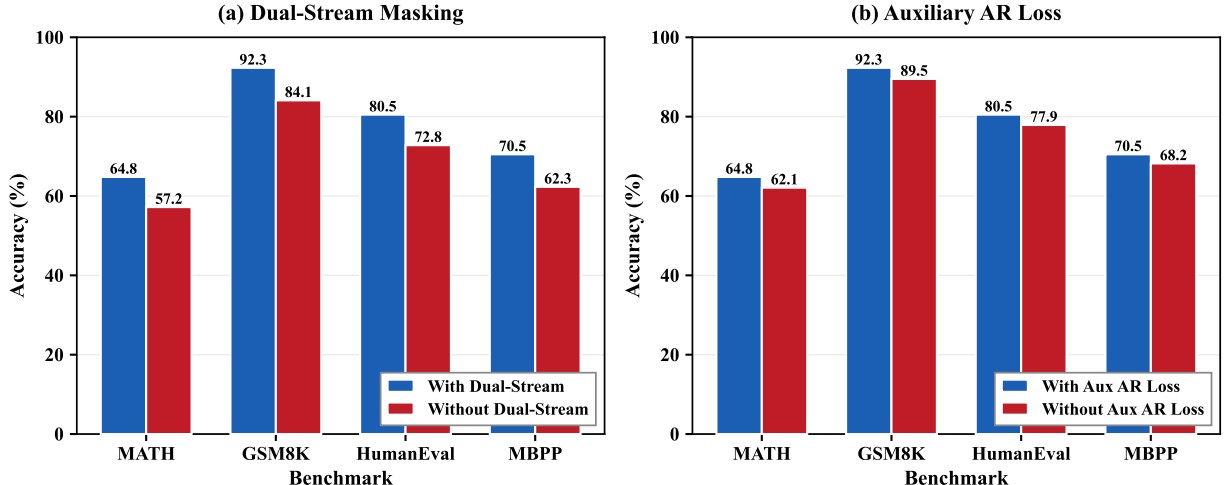

*Figure 6.* **Ablation on key components.** (a) **Dual-Stream Masking**: removing this component causes 5–10 point drops across benchmarks, confirming that simulating inference-time token distribution is critical. (b) **Auxiliary AR Loss**: removing the auxiliary objective leads to 2–3 point degradation, validating its role in preserving autoregressive capabilities.

algorithm operates identically regardless of task domain, and the position-aware confidence mechanism adapts to different entropy distributions without affecting the fundamental caching behavior.

**Baseline models show task-dependent variability.** In contrast, bidirectional models (Dream, LLaDA) and block-wise SDAR exhibit noticeable variation across tasks. On mathematical benchmarks (MATH, GSM8K), these models achieve relatively higher $p_{\text{cache}}$ (0.08–0.13 with KV caching) because mathematical reasoning allows more tokens to be confidently unmasked per diffusion step—arithmetic operations and equation solving produce predictable token sequences. However, on code generation benchmarks (HumanEval, MBPP), $p_{\text{cache}}$ drops significantly (0.04–0.08) because each diffusion step can only unmask fewer tokens in code generation.

**Theoretical upper bound.** The theoretical upper bound for $p_{\text{cache}}$ in current DLLMs is $0.5$, since each token must be processed through the model twice: once as a mask token during generation, and once as an unmasked token to provide context for subsequent positions. WeDLM's achieved values of $0.45$–$0.48$ approach this optimal limit across all tasks, demonstrating that our causal design maximally exploits prefix cacheability.

*Table 5.* Prefix cacheability ($p_{\text{cache}}$) comparison across diffusion models on different benchmarks with block size $B = 8$. Higher values indicate better cache efficiency. WeDLM shows consistent performance across tasks, while baselines exhibit higher $p_{\text{cache}}$ on math and lower on code.

| Model | KV Cache | Math | | Code | |
|---|---|---|---|---|---|
| | | **MATH** | **GSM8K** | **HumanEval** | **MBPP** |
| WeDLM-8B (Ours) | ✓ | **0.47** | **0.48** | **0.45** | **0.46** |
| SDAR-8B | ✓ | 0.11 | 0.12 | 0.07 | 0.08 |
| Dream-7B | ✓ | 0.09 | 0.08 | 0.04 | 0.05 |
| | ✗ | 0.00 | 0.00 | 0.00 | 0.00 |
| LLaDA-8B | ✓ | 0.10 | 0.09 | 0.05 | 0.06 |
| | ✗ | 0.00 | 0.00 | 0.00 | 0.00 |

The task-dependent behavior of baseline models can be explained by the number of tokens unmasked per diffusion step. In mathematical reasoning, structured computations (e.g., "$3 \times 4 = 12$") allow more tokens to reach high confidence simultaneously within a block. In code generation, however, tokens exhibit higher entropy due to variable names, function calls, and syntax variations, resulting in fewer tokens being finalized per step. WeDLM's streaming approach circumvents this issue entirely: tokens are committed individually as soon as they form a contiguous prefix, independent of how many tokens can be unmasked per step.

## F.1. Case Study

To better understand the performance characteristics of `WeDLM`, we analyze its generation behavior across different task modalities. The decoding speed is strongly correlated with the entropy of the output distribution, as shown in the representative cases:

- **Low Entropy (Sequential Patterns):** As shown in Figure 7, the model achieves a peak throughput of **1673.3 tokens/s** on a simple counting task. The deterministic nature of the sequence yields extremely low entropy, allowing the model to speculate and accept many tokens per step.
- **Medium Entropy (Structured Reasoning):** Figure 8 demonstrates a mathematical derivation task. Despite requiring logic, the syntactic structure of the solution is relatively predictable, maintaining a high speed of **745.2 tokens/s**.
- **High Entropy (Open-ended Generation):** In Figure 9, where the model explains Quantum Physics, the speed drops to **197.8 tokens/s**. The high semantic diversity and lexical uncertainty in open-ended text reduce the confidence of speculative tokens, limiting the effective parallel block size.

These results highlight a significant performance disparity: while low-entropy tasks achieve over $8\times$ speedup, high-entropy generation sees diminishing returns. Although this variance partially reflects the intrinsic uncertainty of natural language, it exposes a limitation of the current framework in handling high-perplexity scenarios. Closing this gap—potentially through more robust acceptance mechanisms or dynamic entropy calibration—remains a critical direction for future work to ensure consistent acceleration across all domains.

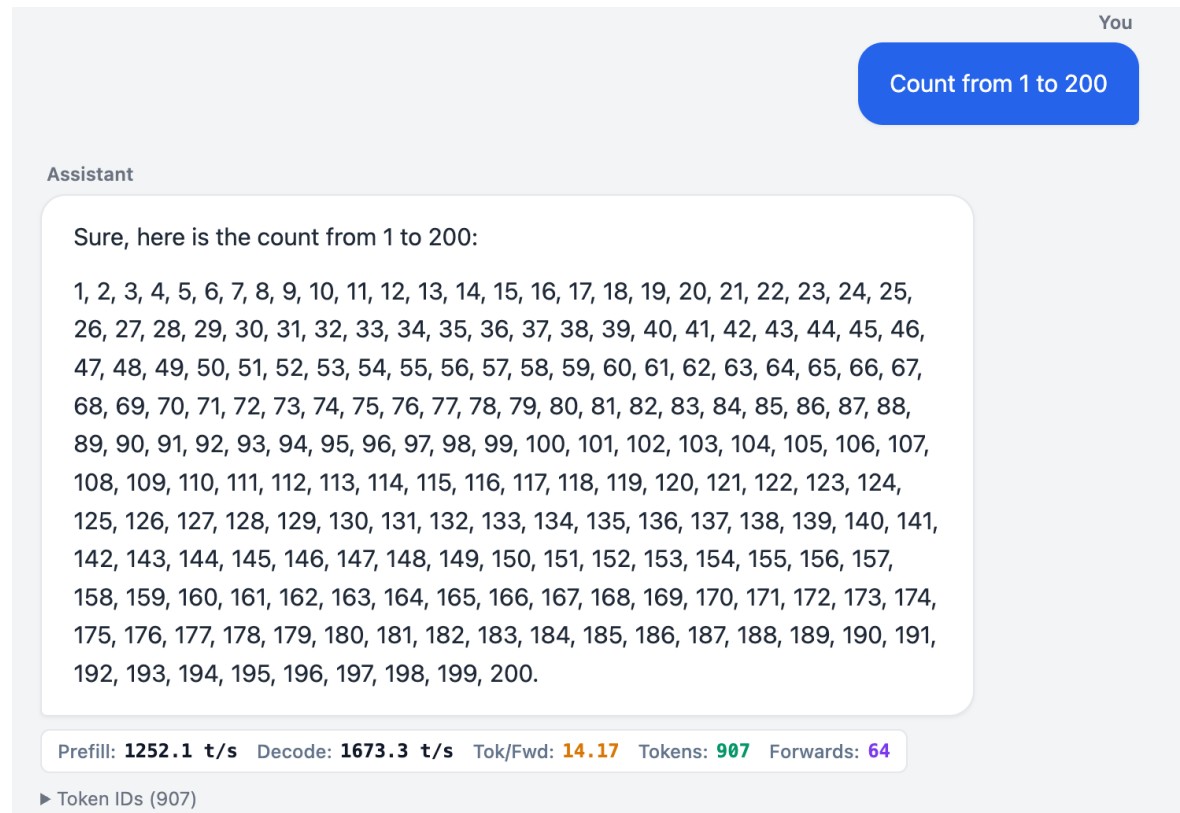

*Figure 7.* **Low Entropy Case:** A simple counting task from 1 to 200. Due to the highly predictable deterministic pattern, `WeDLM` achieves a decoding speed of **1673.3 tokens/s**.

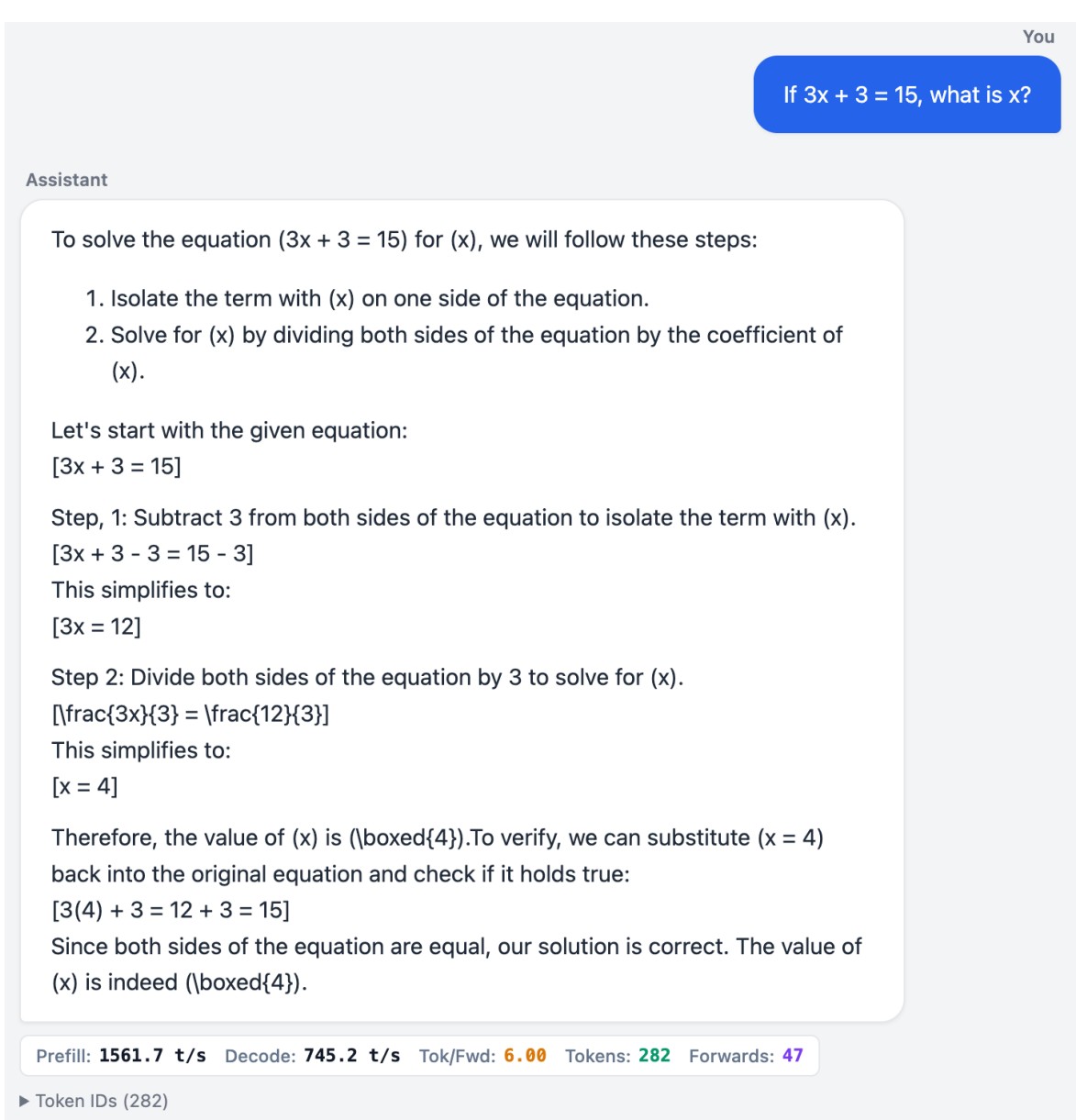

*Figure 8.* **Low Entropy Case:** A mathematical reasoning task solving a linear equation. The structured nature of step-by-step derivation allows significant parallel decoding, achieving **745.2 tokens/s**.

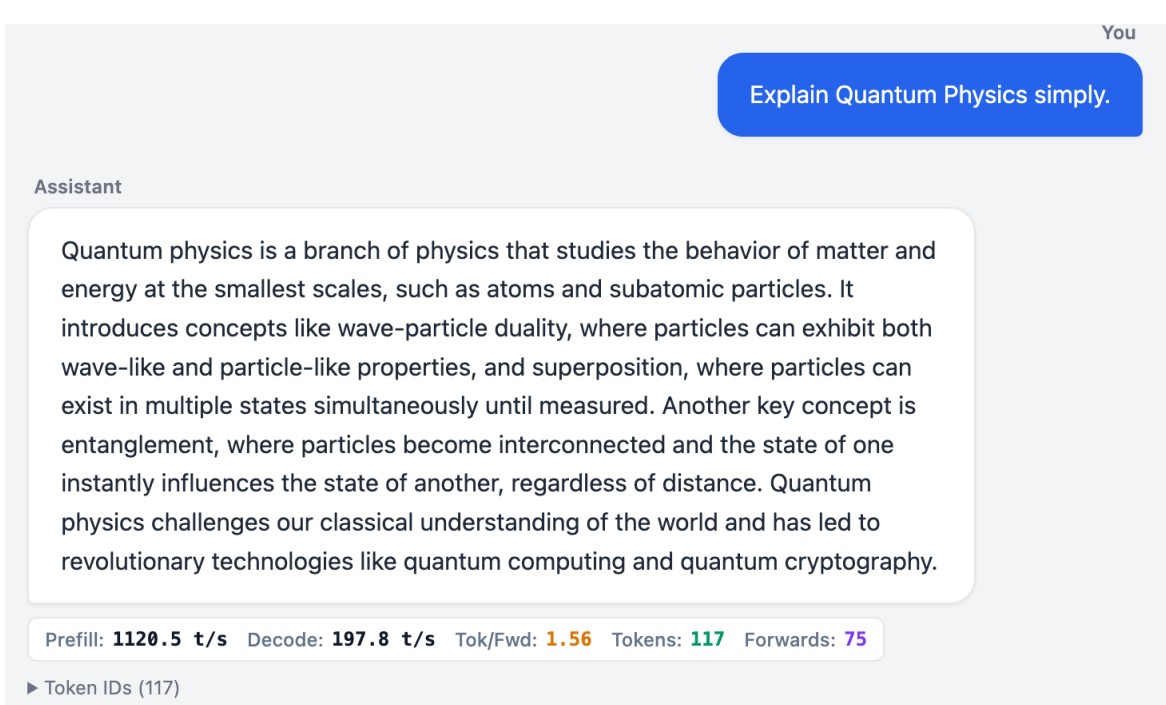

*Figure 9.* **High Entropy Case:** An open-ended knowledge explanation (Quantum Physics). High semantic diversity and precise lexical selection reduce effective parallel block size, resulting in **197.8 tokens/s**.

