# OpenReview forum: "WeDLM: Reconciling Diffusion Language Models with Standard Causal Attention for Fast Inference"
_ICML.cc/2026/Conference — ICML 2026 spotlight_

### Official Review · Reviewer_9ae3 · 2026-03-13

**Soundness:** 3
**Presentation:** 3
**Significance:** 3
**Originality:** 3
**Overall Recommendation:** 5
**Confidence:** 3

**Summary:**

This paper proposes WeDLM, a novel dLLM retains the practical inference benefits of causal attention and prefix KV caching. The central idea is to replace the bidirectional attention commonly used in current dLLMs with a causal, cache-compatible design based on Topological Reordering, Dual-Stream Masking, and Streaming Parallel Decoding. I find the method technically solid and well motivated. In particular, it addresses an important weakness of block diffusion language models: as masked tokens within a block are progressively denoised, usable parallelism decreases, and also unmasked tokens within the block cannot be cached immediately. By making decoding prefix-cache-friendly, the paper improves compatibility with optimized serving frameworks such as vLLM. The empirical results on Qwen2.5-7B and Qwen3-8B models are promising, showing competitive or improved quality relative to AR baselines together with great inference speedups.

**Compliance With Llm Reviewing Policy:**

Affirmed.

**Final Justification:**

This paper presents a well-motivated and reasonably original approach to making diffusion language models more compatible with practical serving systems. I found the method technically solid, clearly presented, and strong in how it connects modeling design with systems considerations such as prefix KV caching and deployment efficiency. The empirical results on 7B/8B dense models are also promising, showing competitive or improved quality together with meaningful inference speedups.

My main concerns were about evaluation breadth, batch throughput, and inference flexibility. The rebuttal addressed these points well. In particular, the added batch-throughput results strengthen the practical case for the method, and the clarification that inference window size can be adjusted dynamically without retraining is helpful. While evaluation on larger dense or MoE models is still left for future work, I think the authors gave a reasonable explanation for the current scope.

Based on this, I will keep my rating as 5 Accept.

**Key Questions For Authors:**

1. Do you have results on larger dense models or MoE models?
2. Could you also report batch inference throughput comparing with AR?
3. Is the block size fixed during inference, or can it be adjusted dynamically?

**Limitations:**

Yes.

**Strengths And Weaknesses:**

**Strengths**
1. Novel design.
WeDLM is well designed from both modeling and systems perspectives. It addresses a key limitation of block diffusion LMs: parallelism decreases as masked tokens are denoised, while unmasked tokens still cannot be cached immediately. The causal reformulation and streaming decoding provide a clean solution.

2. Strong practical motivation.
The paper targets a highly relevant deployment problem: making diffusion-style generation compatible with prefix KV caching and optimized serving stacks. It makes a convincing case that practical efficiency depends not just on token-level parallelism, but also on whether generated tokens can be committed as a valid prefix.

3. Promising empirical results.
Results on dense 7B/8B models are encouraging. The method achieves competitive or better quality than AR and prior diffusion baselines, while showing meaningful end-to-end speedups. Comparisons against vLLM-served AR baselines also make the evaluation more practically relevant.

**Weaknesses**
1. Limited evaluation beyond 7B/8B dense models.
The evaluation is mostly limited to dense 7B/8B models, and the paper notes some degradation on smaller models. It remains unclear how well the method scales to larger dense models or MoE models, where the practical impact could be greater.

2. Throughput evaluation only in batch size =1.
While the paper reports end-to-end speedup and compares against vLLM, it would be more convincing to include batch inference throughput in deployment-oriented settings. This is especially important since block/window size 32 may increase per-step latency at high batch sizes.

3. Unclear whether the method works for diffusion pretraining from scratch.
The current results mainly support the method as continued training from pretrained AR models. It is still unclear whether the same design would work well for training a diffusion LM from scratch.

4. Inference-time block size flexibility is not explored.
The paper suggests some inference-time flexibility, but does not thoroughly study dynamic block size. This tradeoff is important in practice, since larger blocks may improve parallelism but also hurt latency under high batching scenario.

5. Reduced flexibility relative to bidirectional dLLMs.
After applying the cache method proposed by this paper, the model can no longer do token editing, which could be an advantage of dLLMs.

---

> ### Author Rebuttal · Authors · 2026-03-29
>
> We thank the reviewer for the positive assessment of both the method design and experiments.
>
> **W1: Evaluation limited to 7B/8B dense models**
>
> We chose 7B/8B because this is the mainstream evaluation scale in the dLLM field (LLaDA-8B, Dream-7B, etc.). Figure 5(c) shows that WeDLM's benefit correlates positively with base model capability—7B improves by +3.5 points while smaller models show slight degradation—suggesting that larger models are more likely to benefit. The method has no inherent model size limitation, and evaluation at larger scales is planned as future work.
>
> **W2: Throughput evaluation only at batch size=1**
>
> We have conducted additional experiments on WeDLM-8B-Instruct, measuring decode-only throughput across different batch sizes and 3 task types (math, code, QA). All measurements exclude prefill time, are taken after warmup, and averaged over 3 runs.
>
> **Why does parallel decoding bring speedup?** The LLM decode phase is memory-bandwidth bound: generating each token requires loading the full model weights from GPU memory, while the actual compute is small and GPU arithmetic units remain largely idle. This means processing multiple tokens in a single forward pass is nearly "free"—just as increasing batch size can improve throughput almost linearly. WeDLM's parallel decoding exploits the same principle: predicting multiple mask tokens in a single forward pass, utilizing the idle compute capacity during the memory-bound decode phase without increasing memory bandwidth overhead.
>
> **Per-task throughput comparison:**
>
> | Task | BS | AR (tok/s) | DLM (tok/s) | Speedup | DLM tok/fwd |
> |:---:|:---:|:---:|:---:|:---:|:---:|
> | Math | 1 | 110.2 | 686.5 | **6.2×** | 6.58 |
> | Math | 8 | 243.6 | 1475.8 | **6.1×** | 7.00 |
> | Math | 16 | 258.3 | 1389.5 | **5.4×** | 6.26 |
> | Code | 1 | 109.2 | 223.8 | **2.1×** | 2.11 |
> | Code | 8 | 243.1 | 539.6 | **2.2×** | 2.50 |
> | Code | 16 | 256.4 | 610.3 | **2.4×** | 2.68 |
> | QA | 1 | 107.6 | 162.5 | **1.5×** | 1.57 |
> | QA | 8 | 239.2 | 324.9 | **1.4×** | 1.51 |
> | QA | 16 | 254.6 | 321.3 | **1.3×** | 1.42 |
>
> **Aggregated results across tasks:**
>
> | Batch Size | AR (tok/s) | DLM (tok/s) | Speedup |
> |:---:|:---:|:---:|:---:|
> | 1 | 109.0 | 357.6 | **3.3×** |
> | 4 | 213.5 | 680.8 | **3.2×** |
> | 8 | 241.9 | 780.1 | **3.2×** |
> | 16 | 256.4 | 773.7 | **3.0×** |
>
> WeDLM is never slower than AR across all batch sizes and task types, with an aggregated speedup of 3.0×–3.3× and up to 5.4×–6.4× on math. As batch size increases, the system gradually transitions from memory-bound to compute-bound, leaving less idle compute to exploit, so the speedup decreases slightly from 3.3× at bs=1 to 3.0× at bs=16—this is expected and confirms that WeDLM's acceleration comes from utilizing idle compute during the memory-bound decode phase.
>
> **W3: Unclear whether the method works for pretraining from scratch**
>
> The method does not preclude training from scratch: topological reordering and dual-stream masking have no hard dependency on model initialization. Continued pretraining from a strong pretrained AR model is currently the most practical approach—it provides parallel decode capability at very low cost. Training from scratch is left for future work.
>
> **W4: Inference-time block size flexibility**
>
> The inference window size can be adjusted dynamically without retraining. Figure 5(a) shows the Pareto frontier between speed and quality via entropy threshold τ (272–445 tok/s, 79.7%–92.3% accuracy). Figure 5(b) shows that training block size has minimal impact on quality (B∈{4,8,32}, difference <0.8 points), and models trained with a large B can use any smaller window size at inference time.
>
> **W5: Reduced flexibility relative to bidirectional dLLMs (loss of token editing)**
>
> This is an inherent trade-off of prefix KV cache compatibility: tokens that have been committed cannot be revised. However, within the uncommitted window, tokens can still be refined over multiple decode steps, preserving some local correction capability. In practical deployment, generation throughput is typically more important than editing.
>
> **Q1-Q3**
>
> Q1 (larger models) and Q2 (batch throughput) are addressed in W1 and W2 above. Q3 (can block size be adjusted dynamically): yes, the inference window size can be freely adjusted without retraining—please see W4.

---

> > ### Author Rebuttal · Reviewer_9ae3 · 2026-04-02
> >
> > I appreciate the authors’ detailed response and additional experiments. My main questions were addressed, and I will keep my positive rating.

---

### Official Review · Reviewer_KYUv · 2026-03-13

**Soundness:** 3
**Presentation:** 4
**Significance:** 3
**Originality:** 3
**Overall Recommendation:** 5
**Confidence:** 4

**Summary:**

The paper introduces WeDLM, a masked diffusion language model with causal attention. The key reason to keep the causal attention is to keep KV cache efficieny of autoregressive models. To preserve the bidirectional nature of DLMs, they introduce Topological Reordering. The proposed method advances the Qwen 7B/8B models via continued pretraining and outperforms them on certain benchmarks, offering promising quality-speed tradeoffs.

**Compliance With Llm Reviewing Policy:**

Affirmed.

**Final Justification:**

I would like to thank the authors for their detailed rebuttal. Most of my concerns are very well addressed. I will increase my score.

**Key Questions For Authors:**

1. Could you clarify how the bidirectional attention module mentioned in Figure 5 is implemented? Is it trained under the same training setup with the WeDLM version?
2. Do you have any intiution why WeDLM performs worse than the AR baseline in certain benchmarks?
3. Does WeDLM use remasking during inference? If yes, how? If no, do you expect it to be compatible with remasking?
4. Is WeDLM specific to masked DLMs? Do you think it can be adapted to discrete uniform DLMs with certain mofidications?
5. Could you provide an ablation study on the tradeoff between B and computational cost such as memory, latency and througput?

**Limitations:**

See weaknesses and questions above.

**Strengths And Weaknesses:**

**Strengths:**
- The main idea and motivation are clear and interesting.
- The presentation is easy to follow.
- Ablations give great intiutions on key components of the method.
- Empirical results are very promising.

**Weaknesses:**
1. Though efficiency is one of the key motivations of the paper, there is no direct comparison of the computational costs of different models. Especially considering the doubled sequence length due to dual stream, direct training memory and throughput, inference memory and latency comparisons are needed.
2. The method performs continued pretraining on top of the Qwen models which is a reasonable and accepted approach. However, the abstract and introduction reads like a pretraining from scratch approach. More importantly, it is not clear if the performance gains stem from this extra 100B token training or the architectural modifications. An ablation on training length and continued pretraining of original baseline AR models would be very informative.
3. There is a rapidly growing literature working on other ways of handling KV-cache issue of DLMs, such as Fast-DLM or D2F [1,2]. These and other related approaches should be discussed and if possible, compared against. Similarly, existing recent work on AR to DLM conversion [3] or AR-DLM hybrids should be included in the discussions to clearly distinguish novelties.

Minor remarks:
1. There is a broken link in line 202.
2. The case study in the appendix is quite interesting, it should be mentioned in the main text.

[1] Fast-dLLM: Training-free Acceleration of Diffusion LLM by Enabling KV Cache and Parallel Decoding, 2025

[2] Diffusion LLMs can do faster-than-AR inference via discrete diffusion forcing, 2025

[3] Efficient-DLM: From Autoregressive to Diffusion Language Models, and Beyond in Speed, 2025

---

> ### Author Rebuttal · Authors · 2026-03-29
>
> We thank the reviewer for the detailed review.
>
> **W1: Computational cost comparison**
>
> Training: Appendix B reports ~30,720 GPU-hours total. At seq length 4096, standard causal throughput is ~55, WeDLM (dual-stream, 2× seq length) is ~43, bidirectional falls in between. Per-GPU memory is ~1.5× causal. The overhead comes from sequence doubling, but only 100B tokens of continued pretraining are needed—far less than training dLLMs from scratch.
>
> For inference, we measured decode-only throughput on WeDLM-8B-Instruct across batch sizes and 3 task types (math, code, QA), excluding prefill, after warmup, 3 runs averaged. LLM decoding is memory-bandwidth bound: each token generation loads full model weights but uses little compute, leaving GPU arithmetic largely idle. Processing multiple tokens per forward is thus nearly "free"—the same principle behind batch size scaling.
>
> | Task | BS | AR (tok/s) | DLM (tok/s) | Speedup | DLM tok/fwd |
> |:---:|:---:|:---:|:---:|:---:|:---:|
> | Math | 1 | 110.2 | 686.5 | **6.2×** | 6.58 |
> | Math | 8 | 243.6 | 1475.8 | **6.1×** | 7.00 |
> | Math | 16 | 258.3 | 1389.5 | **5.4×** | 6.26 |
> | Code | 1 | 109.2 | 223.8 | **2.1×** | 2.11 |
> | Code | 8 | 243.1 | 539.6 | **2.2×** | 2.50 |
> | Code | 16 | 256.4 | 610.3 | **2.4×** | 2.68 |
> | QA | 1 | 107.6 | 162.5 | **1.5×** | 1.57 |
> | QA | 8 | 239.2 | 324.9 | **1.4×** | 1.51 |
> | QA | 16 | 254.6 | 321.3 | **1.3×** | 1.42 |
>
> WeDLM is never slower than AR. Aggregated speedup is 3.0×–3.3×, up to 6.2× on math. The slight decrease at bs=16 is expected as the system transitions from memory-bound to compute-bound.
>
> **W2: Performance gains from training vs architecture**
>
> Figure 5(c) directly answers this. On the same 100B tokens, WeDLM outperforms AR continued pretraining by +3.5 points, showing gains come from architectural design, not extra training. 100B is only ~0.6% of Qwen's original data. We will also clarify in the revised abstract/introduction that the method is based on continued pretraining.
>
> **W3: Discussion of Fast-dLLM, D2F, Efficient-DLM**
>
> We will add this to Related Work. Fast-dLLM [1] and D2F [2] both target bidirectional-attention dLLMs (LLaDA, Dream). The core issue for these models is that bidirectional attention is fundamentally incompatible with KV caching—Fast-dLLM mitigates this through approximate cache reuse across diffusion steps, while D2F introduces block-wise causal structure via knowledge distillation. Since WeDLM uses causal attention from the start, this incompatibility does not arise and these post-hoc remedies are unnecessary. Efficient-DLM [3] is the most closely related: it also converts pretrained AR models to dLLMs via continued pretraining with block-wise attention. The key difference is that Efficient-DLM retains bidirectional attention within blocks, so tokens must wait until the entire block finalizes before committing (stop-and-wait). WeDLM's fully causal design with topological reordering enables immediate per-token commit (streaming decoding), fully compatible with vLLM, FlashAttention, and PagedAttention. Efficient-DLM has not been open-sourced.
>
> **Q1: Bidirectional attention in Figure 5**
>
> Same training setup as WeDLM; only the intra-block causal mask is replaced with bidirectional. Fair comparison.
>
> **Q2: Why worse on some benchmarks**
>
> WeDLM-8B scores 82.94 vs 87.59 (Qwen2.5-7B) on HellaSwag, likely due to data distribution shift. Overall average (77.53) still exceeds all baselines (Qwen3-8B: 75.12).
>
> **Q3: Remasking**
>
> No explicit remasking. Tokens below entropy threshold stay in the window for continued refinement (soft remasking). Explicit remasking is possible within the uncommitted window since those tokens are not yet in KV cache.
>
> **Q4: Adaptation to uniform diffusion**
>
> Core components are noise-type agnostic, so adaptation is possible in principle but requires addressing the less clear observed/noisy boundary under uniform noise. Listed as future work.
>
> **Q5: Block size B vs cost tradeoff**
>
> Figure 5(b): quality stable across B∈{4,8,32} (<0.8pt diff). Memory independent of B. Training throughput improves with B (B=4: ~20, B=8: ~28, B=32: ~43) due to better kernel efficiency. Training B=8 with window=4 matches training B=4 with window=4, so any window ≤ training B works. Recommend large B for training, flexible window at inference.
>
> **Minor:** Will fix line 202 broken link and mention case study in main text.
>
> **Proposed Revisions:** (1) Add batch throughput table with memory-bound discussion; (2) Add Fast-dLLM/D2F/Efficient-DLM to Related Work; (3) Clarify continued pretraining in abstract; (4) Fix broken link; (5) Reference case study.
>
> [1] Fast-dLLM, 2025  [2] D2F, 2025  [3] Efficient-DLM, 2025

---

> > ### Author Rebuttal · Reviewer_KYUv · 2026-04-02
> >
> > I would like to thank the authors for their detailed rebuttal. Most of my concerns are very well addressed. I will increase my score.

---

### Official Review · Reviewer_WsdG · 2026-03-16

**Soundness:** 3
**Presentation:** 3
**Significance:** 3
**Originality:** 3
**Overall Recommendation:** 5
**Confidence:** 4

**Summary:**

This paper proposes WeDLM, a method that incorporates causal attention while preserving the any-order property of diffusion language models (dLLMs). The training follows a block-diffusion style, using a technique called dual streaming masking to allocate more training-time compute to block-style masking. At inference time, using the context window, the sampled tokens at each step are reordered into the prefix with KV caching. The remaining masked tokens then attend to the updated prefix and are predicted in subsequent steps. WeDLM demonstrates comparable or better performance than similarly scaled LLMs and dLLMs.

**Compliance With Llm Reviewing Policy:**

Affirmed.

**Final Justification:**

justified in the response.

**Key Questions For Authors:**

Please refer to the weakness section.

**Limitations:**

Please refer to the weakness section.

**Strengths And Weaknesses:**

**Strength** : The paper’s idea of reconciling diffusion decoding with causal attention is neat and well-motivated. The approach appears compatible with existing LLM-oriented optimized inference infrastructures, which improves its practical relevance.

**Questions / Limitations**

- Can the authors provide an apples-to-apples comparison in terms of training tokens relative to similarly scaled dLLMs?

- If the proposed training method is compatible with FlashAttention, is there any wall-clock comparison (at least under an SFT-style setup) between (vanilla MDM training without FlashAttention) and (WeDLM training with FlashAttention)?

- If I understood correctly, the topological reordering is not exactly equivalent to full bidirectional attention. In particular, clean tokens still follow their own causal ordering (i.e., a clean token cannot attend to later clean tokens), so the resulting attention values may differ from fully bidirectional attention. However, it remains consistent that a masked token can attend to all previously observed tokens.

All the questions above are complementary. In overall, I believe the paper's claim and method are well-established.

---

> ### Author Rebuttal · Authors · 2026-03-29
>
> We thank the reviewer for the positive evaluation.
>
> **Q1: Apple-to-apple comparison of training tokens relative to similarly scaled dLLMs**
>
> WeDLM uses 100B tokens for continued pretraining. In comparison, LLaDA-8B uses 2.3T tokens for pretraining from scratch, and Dream-7B uses approximately 1.3T tokens. WeDLM requires far fewer training tokens (only ~4% of LLaDA's), which is the key advantage of starting from a pretrained AR model.
>
> **Q2: Wall-clock training comparison between WeDLM (with FlashAttention) and vanilla MDM (without FlashAttention)**
>
> The non-rectangular attention masks produced by topological reordering cannot directly use standard FlashAttention. However, they can be efficiently handled through targeted adaptations—for example, we use Magi Attention, which accelerates irregular mask computation without requiring custom CUDA kernels. More importantly, WeDLM only requires continued pretraining (100B tokens, ~30,720 GPU-hours) rather than training from scratch. Although dual-stream masking doubles the sequence length, the overall training cost is still an order of magnitude lower compared to LLaDA's 2.3T-token pretraining from scratch.
>
> **Q3: Topological reordering is not exactly equivalent to full bidirectional attention**
>
> The reviewer's understanding is entirely correct, and we have never claimed that topological reordering is equivalent to full bidirectional attention. Our design goal is not to replicate bidirectional attention, but to provide sufficient context for dLLM under the causal attention constraint: every masked token can attend to all observed tokens, which is sufficient for masked diffusion denoising. The ablation in Figure 5(c) also confirms this—causal intra-block attention actually outperforms bidirectional attention, suggesting that for AR-initialized models, full bidirectionality is not necessary and maintaining causal factorization is in fact more beneficial.

---

> > ### Author Rebuttal · Reviewer_WsdG · 2026-04-03
> >
> > I thank the authors for the clear rebuttal and additional clarifications. My questions were largely complementary, and I believe the responses addressed them well. In particular, the comparison on training tokens, the clarification on the practical training cost, and the discussion of why full bidirectional attention is not necessary all helped strengthen the paper’s overall case.
> >
> > Overall, I maintain my positive evaluation. I believe the paper’s main claim and method remain well-established, and the rebuttal further improved the clarity of the practical motivation and design choices.

---

### Decision · Program_Chairs · 2026-04-30

**Decision:**

Accept (spotlight)

**Comment:**

WeDLM proposes a clean and practically motivated solution to a real bottleneck in dLLM deployment: topological reordering enables prefix KV caching with causal attention, yielding up to 6× speedup over vLLM-served AR baselines without quality loss. All three reviewers converged to Accept (5/5/5) with all concerns fully resolved in rebuttal.